# A reduced perception of sensory information is linked with elevated boredom in people with and without attention-deficit hyperactivity disorder

Johannes P.-H. Seiler [1] ✉, Jonas Elpelt [2,3], Vsevolod Mashkov[1], Aida Ghobadi[1], Ambika Kapoor[4], Daniel Turner[4], Matthias Kaschube [2,3,7], Oliver Tüscher [4,5,6,7] & Simon Rumpel [1,7]

Our brains have evolved to represent and process sensory information from our environment and use it to guide behavior. The perception of sensory information and subsequent responses, such as boredom, however, vary across situations and individuals, impressively depicted by patients with attentional disorders who show extensive boredom across many situations. Despite these implications, it remains unclear how environmental features and individual traits act together to allow effective transmission of sensory information, and how both factors relate to boredom experience. We present a framework to address this issue, exposing human participants to text stimuli with defined objective information content, while assessing perceived information, boredom and text sentiment. Using information theory to formalize external and internal factors of information transmission, we find that lower information transmission predicts higher boredom. Moreover, individuals with attention deficit hyperactivity disorder show lower information transmission, compared to a control sample. Together, delineating the interaction of sensory information content with individual traits, boredom emerges as a situational consequence of reduced information-decoding, heightened in ADHD.

In order to navigate our external world efficiently, individuals have developed intricate sensory systems to register and process the stream of external stimuli they are facing. In this respect, a fundamental ability of our brain is to segregate the encountered sensory stimulation into meaningful elements and interpret them adequately. Through this process, complex features of stimuli in an individual's environment can be sampled by sensory organs and transformed into effective information about the current sensory surrounding. For the transmission of information from external sensory stimuli to an individual's mind, two components are essential: First, the complexity of the sensory input, determining the potentially available information from a stimulus (*external factor*), and second, an individual's abilities to decode and interpret the stimulus[1] (*internal factor*). For instance, an example stimulus such as a blank sheet of paper would only provide low complexity, limiting any information that could be extracted from a reader. In contrast, a sheet of paper containing printed German poems would provide higher complexity, which however additionally requires adequate decoding abilities from an individual - language knowledge in this case - in order to extract the full information.

In this context, the process of extracting information from sensory stimuli has been identified as a mechanism with crucial implications on an individual's abilities to learn[2,3] and adjust to changing conditions[4–6]. Moreover, cognitive mechanisms to estimate and actively seek novel environmental information allow an efficient exploration of an individual's environment[2,7,8], the expansion of individual knowledge[2,9,10] and the

[1]Institute of Physiology, Focus Program Translational Neurosciences, University Medical Center of the Johannes Gutenberg University Mainz, Mainz, Germany. [2]Frankfurt Institute for Advanced Studies, Frankfurt am Main, Germany. [3]Institute of Computer Science, Goethe University Frankfurt, Frankfurt am Main, Germany. [4]Department of Psychiatry and Psychotherapy, University Medical Center of the Johannes Gutenberg University Mainz, Mainz, Germany. [5]Leibniz Institute for Resilience Research, Mainz, Germany. [6]Department of Psychiatry, Psychotherapy and Psychosomatic Medicine, University Medicine Halle, Martin-Luther University Halle-Wittenberg, Halle, Germany. [7]These authors jointly supervised this work: Matthias Kaschube, Oliver Tüscher, Simon Rumpel. ✉e-mail: johseile@uni-mainz.de

development of creative behavior[11,12]. In complement to curiosity which attracts individuals to seek specific pieces of information[3,13,14], boredom constitutes an independent but essential co-factor of information-seeking, driving individuals away from sources of monotony[6,15–19]. Importantly, boredom is typically elicited by conditions of low sensory information[16,19,20] in which individuals are unable to engage their cognitive resources with the current environment[21–23]. Due to its aversive nature, boredom in consequence pushes individuals to turn away from monotonous stimulation and drives them to search for higher information in order to achieve boredom relief[15,16,18,24–27]. Thus, boredom can be described as a "hunger for information"[6], serving an exploratory function for behavior under physiological conditions[15,21]. However, boredom has also been linked with risk-related and dysfunctional behavior in the context of psychopathologies[28–32]. Especially, patients with attentional deficit hyperactivity disorder (ADHD) have been shown to strongly and chronically experience boredom and its consequences[33–35], reacting impulsively and inadequately to states of low information in order to alleviate the lack of stimulation[36–39].

Can extensive boredom experience, as observed in ADHD, be explained by dysfunctional information-processing? And, which factors determine how efficiently external information is transmitted from a sensory stimulus to an individual? In particular, while the transmission of information from an environmental stimulus to an individual has been proposed to serve as a major factor in the emergence of boredom[16,24,40,41], it remains unclear, why a given environment, with defined stimulation sources, can lead to different levels of boredom across individuals. The positive association of boredom and ADHD symptoms[34,42] hints towards a central role of internal decoding-related features and traits in mediating effective information transmission, providing a potential explanation for inter-individual differences in boredom susceptibility. Although plausible, the mutual effects of external and internal factors of information transmission and their implications on boredom and other sentiments have not yet been quantitatively assessed.

In our study, we address this issue by presenting a psychophysical paradigm, which allows a quantitative description of the amount of information provided by a particular stimulus, while at the same time probing how much of this information is effectively conveyed to an individual[1,43]. This paradigm allows to study the individual transmission of information, by presenting human participants with defined text stimuli that systematically vary in their empirical entropy – a standard measure of objective information content[43–46] – while asking them to rate the perceived information, boredom and additional dimensions of sentiment independently for each text. Referring to classical theories of information processing[1], our framework assumes that external stimuli are inevitably perceived and interpreted relative to internal priors and expectations[47–49] which might vary across individuals, hence affecting a person's ability to decode and retrieve sensory information. To cover these internal aspects affecting information transmission, we assessed different personality traits in the participants. Moreover, we contrasted experimental assessments in participants that reported to be free from an active neuropsychiatric disorder (here referred to as *healthy control participants*) with assessments in participants having a clinically diagnosed attention-deficit hyperactivity disorder (here referred to as *ADHD patients*).

Building on this framework, we investigate the link of information transmission and boredom in four steps: (i) We assess and compare the relationship between perceived information, boredom and text sentiment for different stimuli with defined objective information content. (ii) We quantify the impact of external, entropy-based factors versus internal, trait-based factors on the perception of information. (iii) We explore the link of the individual ability to retrieve external information and subjective boredom experience. (iv) Using our healthy control sample as a reference, we test how information transmission, boredom and text sentiment is affected in a clinical sample of ADHD outpatients.

With this, we establish a framework to assess distinct environmental and psychological factors of information transmission in individual humans and provide a model to quantitatively describe boredom as a consequence of reduced information flow to the brain.

## Methods

The study was approved by the local ethics committee (Ethikkommission der Landesärztekammer Rheinland-Pfalz, processing numbers 2024-17477 and 2018-13164_3). There was no pre-registration of the study. Written informed consent was obtained from all participants of the study.

### Healthy control cohort

A total number of 162 mentally healthy students from the University of Mainz was recruited via an online recruiting system[50] to participate in our study. Exclusion criteria for participants were active psychiatric or neurological disorders as well as insufficient German language skills, both assessed by self-reports (see general information questionnaire below). From the initial sample, 20 participants encountered technical problems during the experiment, leading to lacking or incomplete data in text ratings and exclusion from our analyses. Thus, a final sample of 142 participants without active neuropsychiatric disorders was used for the analyses of this study. The sample predominantly comprised females (79.6%) and had an average age of 22.7 years (for further demographic information see Table 1).

### Experimental procedure

The experiment of the study was conducted on a single day in the facilities of the Mainz Behavioral and Experimental Laboratory (MABELLA). Participants were introduced to the experiment and were asked for their agreement to participate. After providing consent, participants started to work on the experiment implemented in a custom MATLAB® program, presented on computers of the behavioral lab. For the full duration of the experiment, participants were instructed to wear headphones to shield potentially distracting noise.

**Psychometric questionnaires.** First, participants were asked to fill out a battery of different self-report scales in order to assess demographic and psychometric properties, such as personality traits and properties of mental health and resilience. Specifically, participants reported *general information* (gender, age, weight, size, patient history) and characteristics in the Big Five personality dimensions to generate a general trait profile (*BFI-10*: Big Five Inventory[51] covering neuroticism, extraversion, openness, agreeableness, conscientiousness), as well as boredom proneness (*BPS*: Boredom Proneness Scale[52,53]) and state boredom at the start of the experiment (*MSBS*: Multidimensional State Boredom Scale[53,54]). Moreover, we assessed symptoms of mental health issues by indicators of trait anxiety (*STAI-Y*: State Trait Anxiety Inventory[55]) and mental resilience (*BRS*: Brief Resilience Scale[56,57]). All questionnaires were presented in German language.

**Generating text stimuli with defined information content.** We sought to psychophysical probe the perceived information content and sentiment for a set of five different short German texts with different information content, operationalized by varying degrees of complexity. To generate these texts, we first created a root text on the topic of winter using ChatGPT (www.chatgpt.com, version 3.5., prompt: "Create a text on the topic of 'winter' that consists of exactly 100 words. The text should have several repetitions of adjectives, nouns, and verbs."). We manually edited this root text to have a high number of word repetitions and low semantic variety, yielding a hardly complex text (text 1, see Supplementary Table 1). We again used this hardly complex text to generate additional three permuted version with gradually increasing semantic diversity and decreasing number of word repetitions (texts 2–4). A last text variant with a high degree of semantic randomness was generated using ChatGPT (text 5; prompt referring to text 4 as a reference: "Edit the following text so that all verbs, adjectives, and

**Table 1 – | Demographic characteristics of the study cohorts**

|  | Healthy control student cohort (n = 142) | ADHD outpatient cohort (n = 19) |
|---|---|---|
| **Gender** | | |
| Male | 29 (20.4%) | 7 (36.8%) |
| Female | 113 (79.6%) | 12 (63.2%) |
| **Age (years)** | | |
| Mean | 22.7 | 32.7 |
| Standard deviation | 5.0 | 9.5 |
| **Active neuro psychiatric disorder** | | |
| No | 142 (100%) | Pre-diagnosed ADHD |

nouns are replaced by words that have as little connection to each other as possible and cover diverse subject areas. The words should be very different and unrelated."). All texts were manually edited and strictly controlled for syntactic correctness, high congruence in their general meaning as well as for an overall count of 100 words (see Supplementary Table 1). Thus, we kept the format and content of the texts vastly congruent, while only varying their complexity and objective information content.

**Probing the perception of text stimuli.** In order to test how participants perceive text stimuli with different complexity, we presented each participant with a random sequence of all five texts and asked them to rate different dimensions of sentiment after reading each text. Specifically, after reading a given text, participants were asked to rate (i) how informative, (ii) how boring, (iii) how creative, (iv) how pleasant (valence) and (v) how arousing they found each text. All questions were presented in random order after the respective text, offering participants a numerical analog scale ranging from 1 ("not at all") to 7 ("completely"). This procedure resulted in one rating of *subjective information content*, *boredom*, *creativity*, *valence* (syn. affect) and *arousal* for each of the texts and each participant.

**Additional elements of the experiment independent from this study.** The assessment of psychometric traits and text perception in our study was integrated into a larger experiment that each participant underwent. In addition to the psychometric scores, participants completed a battery of different language-based divergent thinking tasks[58,59] as well as an auditory scaling task, rating the pairwise similarities of a set of pulsed white noise stimuli. Moreover, participants completed multiple rounds of a two-alternative forced choice task, where they had to discriminate equivalent pulsed auditory stimuli according to different reinforcement contingencies, also rating boredom and the assumed task rule after each round of the task. Lastly, participants also completed a semantic scaling task, similar to the auditory task, rating the pairwise similarities between a set of words.

These other steps of the experiment are conceptually widely independent from our analyses here. Thus, the data of the above subparts is not considered for the present study, but is used for independent studies on auditory perception and creativity[60].

**Assessing text perception in a cohort of ADHD patients**
To contrast the text perception in healthy control participants against the text perception in people showing an intensified search for stimulation, we recruited a sample of 19 adult outpatients with pre-diagnosed attentional deficit hyperactivity disorder (ADHD). These patients constitute a subset of participants recruited for a larger study on boredom in psychiatric patients. ADHD patients were contacted via the Department for Psychiatry and Psychotherapy Mainz and asked for voluntary participation in the study. Participation in the study was fully independent from their psychiatric treatment. The study was conducted at the behavioral laboratory of the

Neuroimaging Center Mainz (NIC). We screened all participants for any other active psychiatric or neurologic co-morbidities, using self-reports on patient history, verifying that there was no other active mental disorder. The sample of ADHD patients showed an average age of 32.7 years with a higher portion of females (63.2%), thus showing qualitative comparability to our healthy control sample (for further demographic information see Table 1). The ADHD patients were presented with the same set of texts, questions and psychometric questionnaires as the healthy control cohort (see above). This assessment was embedded in a larger experiment in which the patients also underwent a set of language-based divergent thinking tasks, a modified two-alternative forced-choice task with visual stimuli[16] as well as a scaling task to rate the pairwise similarities between a subset of the visual stimuli used in the task (study currently in preparation for publication). Importantly, we instructed participants that all experimental steps were independent from each other and presented the five texts with the respective perceptual questions only after the additional experimental steps in order to avoid interference.

**Statistical analysis**
All analyses were conducted using the MATLAB® statistics and machine learning toolbox (The Mathworks Inc., Natick, Massachusetts, USA, version R2022a) and Python (version 3.8.5; packages: Networkx, scipy, Sentence-Transformers). All data was pseudonymized before analyzing.

**Psychometric questionnaires.** The self-reported data was analyzed by computing the sum score for each questionnaire and subscale. Participants that accidentally skipped single items of a questionnaire or subscale were excluded from the respective analysis. This exclusion explains deviations from the total number of recruited participants/patients and the reported *n* of the respective analysis.

**Estimates of text complexity.** We estimated the complexity of each text stimulus, by applying information theoretic and linguistic measures. Specifically, we quantified the cumulated empirical *entropy* $H_{cum} = \sum H_i$ over all words of each text (only referred to as *entropy* in the text). Here, the entropy $H_i$ of a single word $i$ is computed as $H_i = -\sum_{i=1}^{n_i} f_i \log(f_i)$, where, $n_i$ indicates the number of different unique words in the text up to the current word $i$, and $f_i$ represents the relative frequency of word $i$ among all previous words. This metric of cumulated entropy increases over the text stimuli we designed, supporting our a-priori intentions when modulating complexity in the texts (see Fig. 1B). Thus, we used entropy as a proxy for the general objective information content of a stimulus[16,43–46]. In addition to this statistical measure, we estimated the qualitative variety of each text stimulus by computing the mean semantic distance between all its words. Semantic distance yields a reliable measure for the diversity in the thematic content of a text, and has been used as a metric to estimate associative similarity in language networks[61,62]. We computed semantic distance using a state-of-the-art open-source multilingual sentence embedding model based on RoBERTa architecture (https://huggingface.co/T-Systems-onsite/cross-en-de-roberta-sentence-transformer). Supporting our text design, semantic distance developed linearly with increasing complexity in our texts, showing a high correlation to text entropy (see Fig. 1). Due to this high positive correlation and in order to avoid collinearity, we focused our further analyses on the effect of entropy on text perception, constituting a general measure of objective information content of the text stimuli.

**Correlation analyses.** We conducted multiple correlation analyses to test the associations of the different text ratings with participants' sensitivity to external information (see Methods of linear regression below). For all correlations, we report the Pearson coefficient as well the raw *p* values. For the exploratory test of correlations between the psychophysical text ratings and different psychometric scores, we additionally apply a Bonferroni correction for multiple testing.

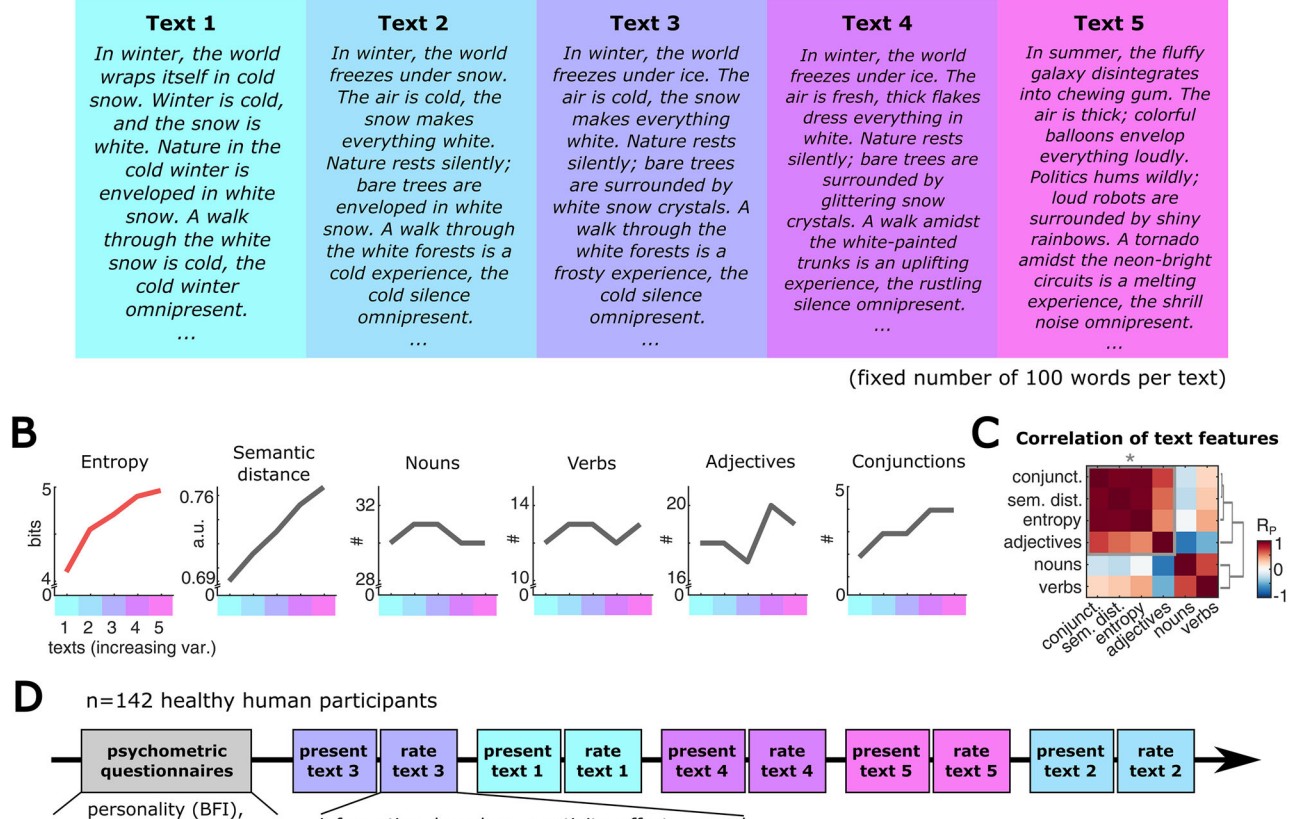

**Fig. 1 | Estimating the conveyed information of text stimuli with defined information content. A** We created a set of five text stimuli systematically varying in repetitiveness and semantic variability (see Methods and Supplementary Table 1; representative text fragments are shown). The topic of the text was fixed, describing a winter scene, except for text 5, containing vastly random, widespread themes. The length of all texts was constrained to 100 words (texts are detailed in Supplementary Table 1). **B** Language features of all texts, illustrating the predefined gradient of objective information content, while having a widely constant format in respect to the word type counts. **C** Correlations of language features (across $n = 5$ texts), indicating positive links between different measures of linguistic complexity (entropy, semantic distance, number of conjunctions and adjectives; highlighted by grey box; *: pairwise correlations of entropy measures are significantly different from zero in a two-sample t-test, $p = 0.022$). **D** Flow of experiment, where 142 healthy participants rated the texts in respect to subjective information content, boredom, creativity, affect and arousal. All texts were presented in random order. In addition, participants completed various self-report questionnaires (general information, BFI-10, STAI-Y, BRS, BPS, MSBS, see Methods).

**Multiple linear regression analysis of information ratings.** To test the dependency of individually perceived information on the amount of objective external information content as well as on internal psychological factors, we conducted a linear regression analysis. Specifically, we fitted each participant's set of information ratings over all five text stimuli with a linear combination of the entropy values of all texts as well as the scores on the Big Five trait questionnaire (openness, agreeableness, extraversion, neuroticism and conscientiousness) as constants. This set of six regressors can be grouped into two conceptually distinct clusters: On the one side, entropy represents *external* stimulus-related factors that affect the individually perceived information, whereas the Big Five traits on the other side reflect *internal* stimulus-independent factors that affect perceived information. For the regression, we used the MATLAB function *fitrlinear* without an intercept. To assess the goodness of the fit, we opposed the mean squared error (MSE) of the regression on the true empirical data with the error of an equivalent regression computed over randomly shuffled data (entropy and trait constants shuffled across texts). We then compared the error distributions between the real versus shuffled data with a Wilcoxon rank sum test. In order to compare the impact of external and internal factors on perceived information, we computed the contribution of each regression parameter $j$ for each participant and text, calculated as $contribution_j = |x_j \beta_j|$, where $x_j$ denotes the respective parameter score for a given text, and $\beta_j$ indicates the fitted

parameter weight over all texts. We next normalized this contribution score to the sum of all parameter contributions, yielding a relative contribution value in the range between 0 and 1. For each of the regression parameters a higher contribution value indicates a stronger effect on the subjective perception of information. Conceptually, the contribution of entropy can be interpreted as a person's sensitivity to perceive and extract information from a stimulus, whereas the contributions of the Big Five parameters in sum represent internal factors that affect perceived information. To compare internal and external factor contributions over all text stimuli, we opposed the entropy contribution and the summed Big Five contributions (note that internal and external factor contributions always add up to a value of one). In our further analyses, we used the mean relative contribution of entropy as an index that describes individual sensitivity to external information (referred to as *information sensitivity* in this study), comparing it between healthy control participants and ADHD patients.

**Comparison of healthy control participants and ADHD patients.** To compare the psychometric profiles of healthy control participants with ADHD patients, we conducted Wilcoxon rank sum tests. Moreover, we tested for differences in the text ratings by applying a Wilcoxon rank sum test for each text rating and reporting the raw $p$-value. To compare these difference effects across all rated sentiments, we computed Cohen's d as a

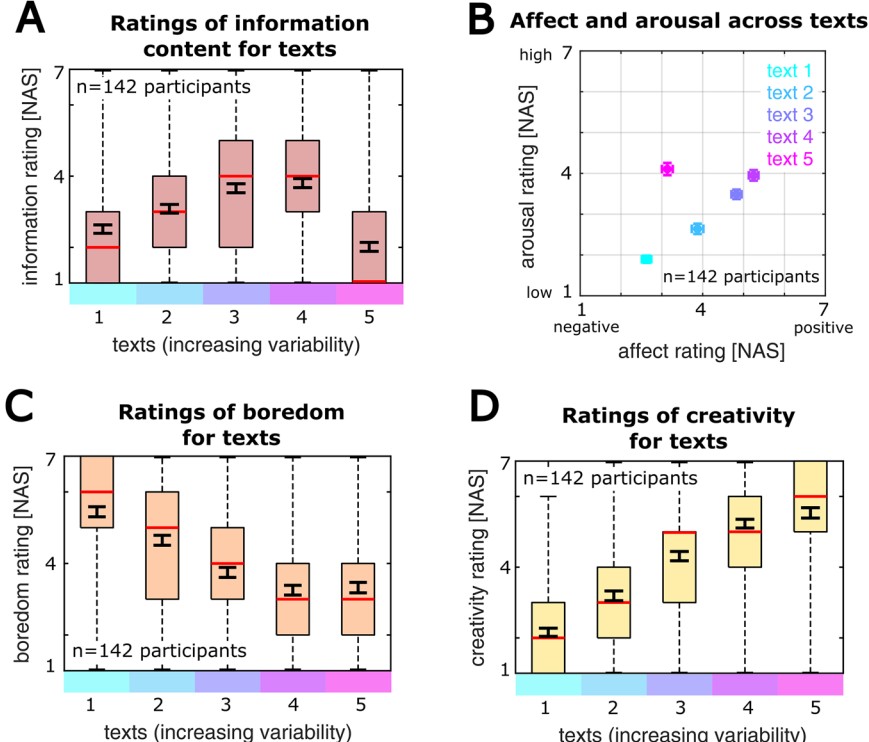

**Fig. 2 | Text perception varies with the objective information content. A** Ratings of subjectively perceived information over all texts ($n = 148$ healthy controls). Ratings follow an inverse U-shaped pattern with maximal information perceived from texts with medium variability. **B** Projection of all texts according to ratings of affect and arousal. Dots indicate the mean, bars indicate the SEM. **C** Ratings of boredom elicited by each text, showing a saturation for text 4-5. **D** Ratings of text creativity, increasing linearly with the variability of the texts. For all boxplots, black indicates the mean +-SEM, red indicates the median, the box indicates the quartiles around the median, and the whiskers indicate the top and bottom quartiles.

measure of effect size, assessing the difference in the rating distributions between healthy control participants and ADHD for each text. These effect size values for the five texts per sentiment were then compared against each other using a Wilcoxon rank sum test. We further compared the sensitivity to external information to the ratings of conveyed information between healthy control participants and ADHD patients, computing the mean relative contribution of entropy over all participants for each text independently, and comparing this vector of mean values between both groups, using a Wilcoxon signed-rank test. All statistical tests, if not explicitly declared differently, were conducted in a two-sided fashion.

## Results

### Establishing text stimuli with defined objective information content

In order to study the individual ability to extract and perceive information from stimuli with defined objective information content, we conducted a psychophysical study with 142 human participants that were free from active neuropsychiatric disorder (demographic characteristics detailed in Table 1). For this study, we first devised a stimulus set of five different German texts that vary in their statistical and semantic complexity (Fig. 1A, Supplementary Table 1). Specifically, we manually constrained the text stimuli to an equivalent word count and theme, while systematically varying their empirical entropy – a standard measure of information content[43–46] – and semantic variety (text 1 with low entropy and semantic variety, text 5 with high entropy and semantic randomness; see Methods). Thus, we obtained a stimulus set with widely constant format and word statistics, primarily varying in measures of objective information content (Fig. 1B). In accordance with this, we found that different, independent measures of text complexity (entropy, semantic

distance, the number of conjunctions and adjectives) showed a positive correlation across the five texts, supporting the validity of our stimulus design (Fig. 1C; mean pairwise Pearson correlation over all combinations of complexity measures across all 5 texts: $n = 6$ pairwise correlations, $\bar{R}_{P.} = 0.488 \pm \text{SD} = 0.219$; Two-sample t-test of the six pairwise correlations of complexity measures against zero: $p = 0.022$). Thus, we establish a textual stimulus set with comparable structure and varying, quantitatively defined objective information content.

### The perception of text stimuli systematically follows the objective information content

We next investigated the general relationship between objective information content, perceived information and text sentiment. For this purpose, we applied a set of standard psychometric questionnaires, and then presented all participants with our five text stimuli, asking them to rate the subjectively perceived *information content*, *affect* (syn. *valence*), *arousal*, *boredom* and the degree of *creativity* for each text (Methods, see Fig. 1D). The information rating was used to estimate the effectively conveyed information between stimulus and receiver. The boredom and creativity ratings were used to assess the cognitive responses to and the appraisal of the stimuli, whereas the affect and arousal ratings characterized text sentiment in classic dimensions of emotion theory[63–65].

Interestingly, we observed that the perceived information content exhibits an inverse U-shaped dependency on text complexity (Fig. 2A; Kruskal-Wallis test over texts: $n = 142$ participants, $p < 0.001$): Texts with low and high randomness were equivalently rated as hardly informative, suggesting that the effectively transmitted information is highest for stimuli with moderate complexity[3,21,66].

Furthermore, we observed a gradual increase of perceived affect and arousal for the text stimuli 1-4 (Fig. 2B; Pearson correlation of mean affect

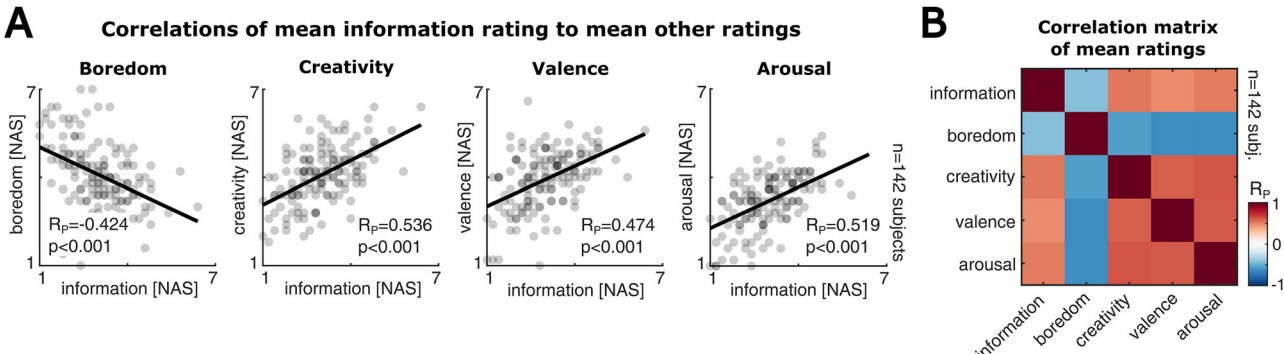

**Fig. 3 | Perceived information is associated with other dimensions of text sentiment. A** Scatter plots and Pearson correlations between participants' mean information ratings over all texts and their mean ratings of boredom, creativity, valence (affect) and arousal. The black line indicates the linear fit. Participants that perceive lower average information in the texts also tend to be more bored by the texts. Moreover, participants with high information rating evaluate the texts as more creative, more positive and more arousing. **B** Pearson correlation matrix of the mean ratings over texts ($p < 0.001$ for all correlations). In line with classical definitions of boredom[69] and the correlations in A, boredom is negatively correlated with creativity, valence and arousal.

and arousal ratings for texts 1-4: $R_P = 0.992$, $p = 0.008$, 95%-CI = [0.879 0.999]), only aborted by the highly random text 5 which was perceived negatively and strongly arousing (Fig. 2B; mean ± SD for text 4 / 5: affect: 5.245 ± 0.116 / 3.134 ± 0.132, arousal: 3.951 ± 0.129 / 4.106 ± 0.149; Wilcoxon rank sum test comparing text 4 and 5: $n = 142$ participants, affect: $p < 0.001$, arousal: $p = 0.294$). In contrast to the trend of increasing perceived information, the boredom ratings declined with increasing text complexity (Fig. 2C; Kruskal-Wallis test over texts: $n = 142$ participants, $p < 0.001$). The highly random text 5 was rated with low boredom, despite exhibiting low ratings of perceived information. These observations suggest that low perceived information from stimuli is associated with a tendency to develop boredom. Lastly, we observed linearly increasing creativity ratings with incrementing text variability, in line with standard frameworks of creativity, operationalizing it by the semantic distance between verbal associations[59,61] (Fig. 2D; Kruskal-Wallis test over texts: $n = 142$ participants, $p < 0.001$; Pearson correlation of mean creativity ratings and text indices: $n = 5$ texts, $R_P = 0.984$, $p = 0.002$, 95%-CI = [0.772 0.999]).

Together, the psychophysical ratings of our study indicate a complex relationship between the objective information of a stimulus and its sentiment: For low to moderate degrees of text complexity, perceived information, boredom, affect and arousal develop widely linearly, whereas very high randomness is characterized by low perceived information, negative valence and high excitement.

**Perceived information is associated with multiple dimensions of text sentiment**

In a next step, we aimed to compare the different dimensions of text sentiment with each other. In particular, we were interested to test the associations between perceived information and the ratings of boredom, affect, arousal and creativity (Fig. 3A). For this, we computed the linear correlation between the participants' mean information rating over all texts and the mean rating of boredom, creativity, affect and arousal respectively (Methods). We found a robust negative correlation of mean perceived information and boredom ($n = 142$ participants, $R_P = -0.424$, $p < 0.001$, 95%-CI = [−0.550 −0.279]), indicating that persons who perceive less information from the stimuli tend to develop more boredom. In contrast, creativity, valence (affect) and arousal ratings showed positive correlations with perceived information ($n = 142$ participants, creativity: $R_P = 0.536$, $p < 0.001$, 95%-CI = [0.407 0.644]; valence: $R_P = 0.474$, $p < 0.001$, 95%-CI = [0.335 0.592]; arousal: $R_P = 0.519$, $p < 0.001$, 95%-CI = [0.387 0.630]), suggesting that participants perceiving more information exhibit higher excitement and more positive affect. These correlation patterns were consistent across all pairwise correlations of the mean ratings (Fig. 3B; $p < 0.001$ for all correlations at a Bonferroni-corrected significance threshold of 0.005). In addition, we observed analogous correlation patterns when repeating this analysis specifically for each text (Supplementary Fig. 1A).

Importantly, we found no significant evidence for an interaction between the text ratings and potentially confounding psychometric features of the participants (Supplementary Fig. 1B; $n = 142$ participants, 44 out of 45 pairwise Pearson correlations indicating no statistical significance with $p > 0.055$, corresponding to the Bonferroni-corrected significance threshold). Merely, the mean boredom rating was positively associated with self-reported state boredom during the experiment, corroborating the validity of the self-reports ($n = 142$ participants, $R_P = 0.269$, $p = 0.001$, 95%-CI = [0.109 0.415], see Supplementary Fig. 1B). Thus, the different text ratings show coherent relations to each other.

**A framework to quantify individually perceived information**

Based on the defined information content of our text stimuli and the ratings of perceived information, we next sought to develop a quantitative model of information transmission in individual participants. For this, we utilized a simplified version of the Shannon-Weaver model[1,43], stating that an environmental source sends out a stimulus with a given objective information content (*external factor* of conveyed information) which needs to be decoded in order to receive its sensory information. This decoding ability itself depends on internal factors, such as personality traits, determining how efficient the objective information can be retrieved (*internal factors* of conveyed information).

We applied this framework to our dataset by conducting a multiple linear regression, fitting the information ratings of each participant with text entropy (a measure for external factors of conveyed information) and the individual Big Five personality characteristics (a measure for internal factors of conveyed information) (see Methods). To be able to interpret our model of information transmission in the context of boredom, we intentionally selected general personality features for our set of internal factors, which did not have a direct link to boredom proneness. We found that the regression model trained on our empirical data outperformed an equivalent control model trained on shuffled data (Fig. 4C; Methods; mean ± SD: real: 1.560 ± 0.121, shuffled: 15.398 ± 3.336; Wilcoxon rank sum test real vs. shuffled: $n = 142$ participants, $p < 0.001$). This shows that a significant and substantial amount of variance in perceived information can already be predicted from mere stimulus entropy and the personality traits included in our model (full regression model: $R^2 = 0.24$).

To assess and compare the relative impact of internal versus external factors on perceived information, we computed the contributions of all parameters in our regression model (Methods). Comparing the pooled internal factor contributions (all Big Five dimensions) and external factor contributions (entropy) across all text stimuli, we found that the relative

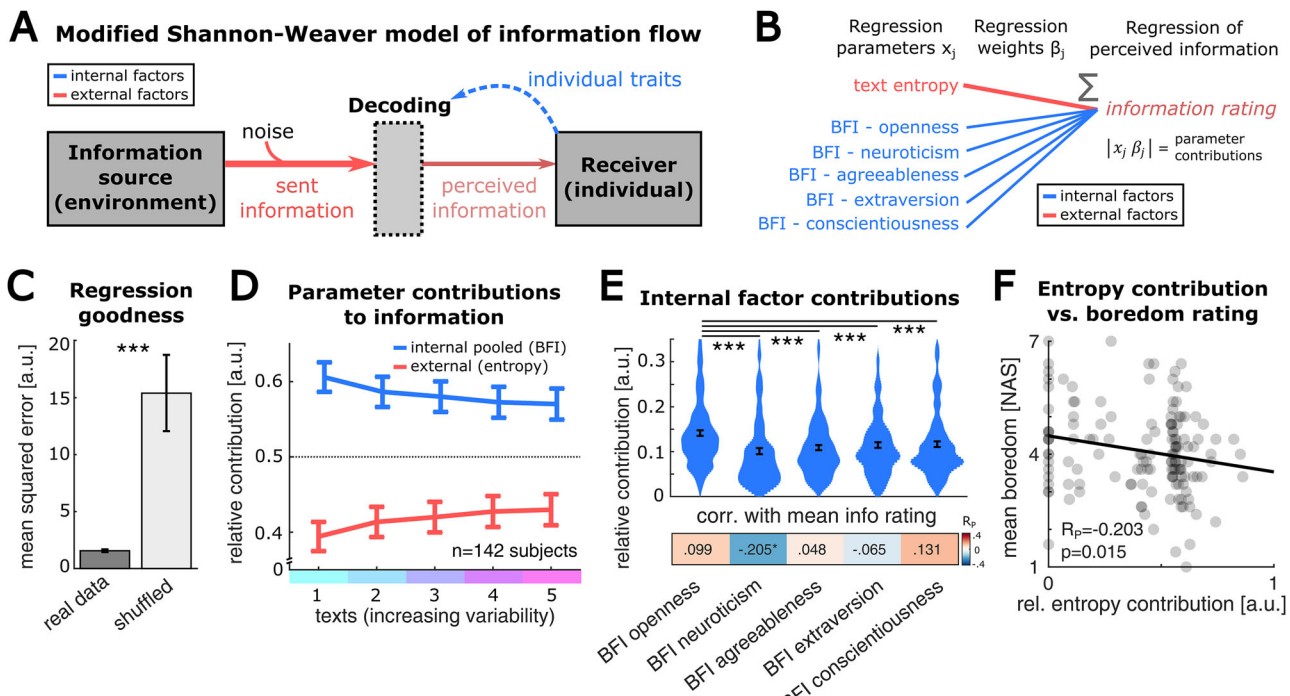

**Fig. 4 | Assessing external and internal factors that determine subjectively perceived information. A** Schematic of a modified Shannon-Weaver model of information flow[1,43]. Information is carried out by an external source (text stimuli) and needs to be decoded by a receiver (participant), determining the amount of information perceived. Decoding abilities, in turn, depend on individual characteristics and traits. Thus, *external factors* (transmitted information by the environment) and *internal factors* (decoding-related traits) together determine the amount of perceived information. **B** Based on the model in (**A**), we assess individually perceived information with a multiple linear regression. For each participant ($n = 142$) we estimate the information ratings of all texts from (i) text entropy (external factor) and (ii) personality traits measured by the Big Five Inventory (see Methods). **C** Error distribution from a regression with the real data vs. a regression with shuffled data ($n = 142$ participants, ***$p < 0.001$ in a Wilcoxon rank sum test). Black bars indicate the mean +-SEM. Gray dots indicate single participant fits. **D** Mean relative

contributions (see Methods) of the external (entropy, in red) and internal (BFI parameters pooled, in blue) drivers of perceived information over all texts ($n = 142$ participants, bars indicate SEM). For low complexity stimuli, internal factors increasingly affect perceived information, whereas for highly complex stimuli entropy dominates perceived information. **E** Top: Distributions of the different internal factors contributions, namely the different Big Five personality dimensions (see Methods, $n = 142$ participants, black bars indicate the mean +-SEM, ***$p < 0.001$ in Wilcoxon rank sum test, all other pairwise comparisons with $p > 0.05$). Bottom: Pearson correlations between BFI scores and mean information rating over all texts (*$p < 0.05$). **F** Scatter plot and Pearson correlation of each participant's mean contribution of entropy (reflecting the individual sensitivity to perceive external information) and the mean boredom rating over all texts ($n = 142$ participants, black line indicates a linear fit).

impact of external factors on perceived information increases with growing stimulus complexity (Fig. 4D; Pearson correlation of mean contributions and text indices: $n = 5$ texts, $R_P = 0.943$, $p < 0.001$, 95%-CI = [0.361 0.996]). This suggests that under conditions of low environmental complexity perceived information depends mostly on internal features, whereas perceived information becomes increasingly affected by external factors in highly complex environments.

Next, we tested the contribution of the specific personality dimensions on perceived information. We observed that *trait openness* had the highest impact on predicting information transmission, whereas the other Big Five dimensions showed comparably lower contributions (Fig. 4E top; mean $\pm$ SD: openness: $0.141 \pm 0.006$, neuroticism: $0.101 \pm 0.007$, agreeableness: $0.109 \pm 0.006$, extraversion: $0.115 \pm 0.006$, conscientiousness: $0.117 \pm 0.006$; Wilcoxon rank sum test openness vs. other dimensions: $n = 142$ participants, $p < 001$ for all tests). Moreover, in order to assess the directionality of impact of the Big Five traits and perceived information, we computed the Pearson correlation of all Big Five scores and the mean information rating (Fig. 4E bottom). We found that neuroticism had a negative effect on information transmission, whereas openness and conscientiousness showed a positive, but non-significant correlation ($n = 142$ participants, openness: $R_P = 0.099$, $p = 0.241$, 95%-CI = $[-0.067\ 0.259]$; neuroticism: $R_P = -0.205$, $p = 0.015$, 95%-CI = $[-0.358\ -0.042]$; agreeableness: $R_P = 0.048$, $p = 0.568$, 95%-CI = $[0.445\ 0.670]$; extraversion: $R_P = -0.065$, $p = 0.241$, 95%-CI = $[-0.227\ 0.101]$; conscientiousness: $R_P = 0.131$, $p = 0.122$, 95%-CI = $[-0.034\ 0.289]$).

Together, this analysis indicates that individually perceived information is to similar extent shaped by external features on the one side, determining the available amount of sensory information, as well as by internal features on the other side, reflecting a person's tendency to cognitively incorporate novel environmental information.

### Low sensitivity to external information is associated with boredom experience

Leveraging our framework of information transmission, we aimed to test the link between individual patterns of perceived information and text sentiment. Therefore, we computed the mean contribution of entropy on perceived information for each participant (Methods), providing a metric of the individual sensitivity to perceive external information (*information sensitivity*). We then correlated this metric with the participants' mean text ratings. We found no credible evidence that creativity was associated with information sensitivity (Supplementary Fig. 2A; $n = 142$ participants, $R_P = 0.033$, $p = 0.698$, 95%-CI = $[-0.132\ 0.197]$), whereas valence and arousal showed a weak positive relationship with information sensitivity (Supplementary Fig. 2B, C; $n = 142$ participants, affect vs. entropy contribution: $R_P = 0.148$, $p = 0.078$, 95%-CI = $[-0.017\ 0.305]$; arousal vs. entropy contribution: $R_P = 0.172$, $p = 0.041$, 95%-CI = $[0.007\ 0.327]$). Interestingly however, we observed a significant negative association of information sensitivity with reported boredom (Fig. 4F; $n = 142$

participants, $R_P = -0.203$, $p = 0.015$, 95%-CI = $[-0.356\ -0.040]$). This correlation is particularly noteworthy, as we did not include a directly boredom-related trait in our model, suggesting that boredom experience is linked to a reduced retrieval of the sensory information provided by the current environment.

## Patients with attention deficit hyperactivity disorder show a reduced sensitivity to external information

Finally, we sought to characterize the perception of information in individuals with a strongly elevated drive to seek environmental information. For this purpose, we recruited 19 adult outpatients with attentional deficit hyperactivity disorder (ADHD) via a specialized outpatient psychiatric department (Methods, demographic information detailed in Table 1), a disorder typically associated with extensive sensation-seeking behavior[36,67]. Although, it is well established that ADHD patients exhibit elevated boredom proneness[34,35], the cognitive causes for this increased boredom experience remain unclear. To address this aspect, the patient cohort was presented with the same set of text stimuli and psychometric questionnaires, rating subjectively perceived information, boredom, creativity, affect and arousal for each text.

As expected, when comparing the psychometric questionnaire scores, we found a significant disparity of ADHD patients and healthy control participants: ADHD patients reported higher boredom proneness and anxiety, and showed lower levels of mental resilience (Supplementary Fig. 3; Wilcoxon rank sum tests comparing the psychometric data of 142 control participants and 19 ADHD patients: BPS/ STAI-Y/ BFI-Neuroticism/ BRS: $p = {<}0.001/ < 0.001/ 0.002/ < 0.001$), illustrating the higher mental burden in the patient group.

We then compared the ratings of perceived information between the healthy controls and the ADHD group. This analysis revealed substantially reduced levels of perceived information in the ADHD patients, especially in conditions of moderate text complexity (Fig. 5A; mean$n$ ± SD for text 1-5: ADHD: 2.00 ± 0.35/ 2.16 ± 0.29/ 2.16 ± 0.22/ 2.53 ± 0.22/ 1.42 ± 0.23, control participants: 2.51 ± 0.12/ 3.08 ± 0.12/ 3.65 ± 0.13/ 3.80 ± 0.13/ 2.01 ± 0.13; Wilcoxon rank sum test for text 1-5: $n = 19$ ADHD patients vs. 142 control participants, $p = 0.053/ 0.006/ < 0.001/ < 0.001/ 0.063$). ADHD patients also tended to report higher boredom for the texts (Supplementary Fig. 4A; mean$n$ ± SD for text 1-5: ADHD: 6.47 ± 0.12/ 4.95 ± 0.42/ 4.26 ± 0.39/ 4.00 ± 0.37/ 3.37 ± 0.37, control participants: 5.44 ± 0.14/ 4.65 ± 0.14/ 3.77 ± 0.14/ 3.26 ± 0.14/ 3.32 ± 0.15; Wilcoxon rank sum test for text 1-5: $n = 19$ ADHD patients vs. 142 control participants, $p = 0.018/ 0.474/ 0.225/ 0.041/ 0.664$), but did not significantly differ from the healthy collective in their ratings of creativity, affect and arousal (Supplementary Fig. 4B–D; Wilcoxon rank sum test for text 1-5: $n = 19$ ADHD patients vs. 142 control participants, creativity: $p = 0.519/ 0.913/ 0.199/ 0.052/ 0.586$, affect: $p = 0.448/ 0.494/ 0.130/ 0.069/ 0.554$, arousal: $p = 0.156/ 0.923/ 0.188/ 0.055/ 0.569$). Comparing the sizes of the difference effects between healthy control participants and ADHD patients across all text sentiments, demonstrated a predominant impairment of perceived information in ADHD, with only minor impact on other dimensions of text sentiment (Fig. 5B; mean Cohen's d ±SD information: 0.659 ± 0.127, boredom: 0.318 ± 0.106, creativity: 0.212 ± 0.098, affect: 0.354 ± 0.089, arousal: 0.242 ± 0.080; Wilcoxon rank sum test information vs. boredom/ creativity/ affect/ arousal: $n = 5$ texts, $p = 0.008/ 0.032/ 0.222/ 0.032$).

Lastly, we applied our framework to assess information transmission to quantify each patient's sensitivity to the external information of the text stimuli. Initially, we tested the link between information sensitivity (expressed by the contribution of entropy in the regression, see Methods) and boredom ratings within our small sample of ADHD patients, finding no significant correlation ($n = 19$ patients, $R_P = 0.299$, $p = 0.214$, 95%-CI = $[-0.179\ 0.663]$) Interestingly, however, we observed that ADHD patients exhibited a consistently reduced information sensitivity throughout all five

text stimuli, as compared with the control participants from our first study (Fig. 5C; mean ± SD: healthy controls: 0.417 ± 0.006, ADHD: 0.382 ± 0.006; Wilcoxon signed rank test comparing the mean entropy contributions over $n = 5$ texts: $p = 0.031$). Together, our analyses demonstrate that ADHD patients show a reduced sensitivity to the information provided by external stimuli, hinting towards a lower ability to decode sensory inputs, predisposing these patients for higher boredom susceptibility (Fig. 5D).

## Discussion

In our study, we investigated the perception of information in defined environmental conditions and the link of information transmission to individual boredom experience and text sentiment. We controlled the objective information content of different text stimuli and obtained individual ratings of the perceived information content and sentiment from healthy control participants and adult ADHD patients. To quantify the individual sensitivities to perceiving environmental information, we established a modified model of information transmission, observing that stimulus entropy as well as individual traits significantly affect the perception of information. Interestingly, lower capabilities to perceive external information are associated with higher boredom experience in our healthy control cohort. Furthermore, ADHD patients show a reduced sensitivity to external information. Together, our study provides evidence that the effective transmission of sensory information to an individual depends on external and individual features, where internal trait-associated decoding capabilities constitute a central driver of boredom in both, healthy control and ADHD samples.

### Information transmission and boredom

A central hypothesis in our study is that information transmission is associated with individual boredom experience. Supporting this assumption, we observed that the perceived information content of a stimulus negatively correlates with the degree of boredom elicited by this stimulus. This finding integrates into previous theories suggesting that boredom serves as a safeguard mechanism to optimize the acquisition and processing of information in the brain[6,16,24,40,41]. In this context, it is assumed that boredom can be elicited by under-stimulating environments on the one side, but also by over-complex, confusing environments which prevent sufficient engagement of the own priors and hence both reduce information flow[21,66,68]. This perspective aligns with previous definitions of boredom characterizing it in respect to a lack of attention and meaning[23,69]. In particular, it was suggested that under-complex as well as over-complex environments increase the difficulty to find meaning in the incoming stimulation, thus leading to boredom[40,41,69]. Despite a general synergy between the meaning-based accounts of boredom and our information-based account in this study, there are also some noteworthy discrepancies: While prior studies on boredom and complexity have focused on manipulating the difficulty of behavioral tasks that participants had to solve with cognitive skills[69,70], we used a psychophysical approach allowing us to assess information transmission through the perception of passively presented stimuli. Thus, our approach focuses on perceptual aspects of information transmission which are widely independent from specific cognitive skills. Furthermore, while previous studies described the complexity of a situation or environment in a qualitative or ordinally scaled manner, our approach uses metrics from information theory to parametrically describe the complexity provided by a stimulus, in in the form of empirical entropy[6,16]. Hereby, our framework quantifies primarily the amount of information which is provided by a given stimulus, not its content. However, a stimulus that carries meaning for an individual would not only need to carry enough information, but also it would need to carry information of a specific topic that is useful for an individual[13]. Amount and content of information together may therefore interact to affect the degree of boredom elicited in an individual.

In the framework of our study, we capture two aspects of transmitted information: On the one side external features relating to the complexity and potential information content of a stimulus, and on the other side, internal features that allow an individual to extract information from a given sensory

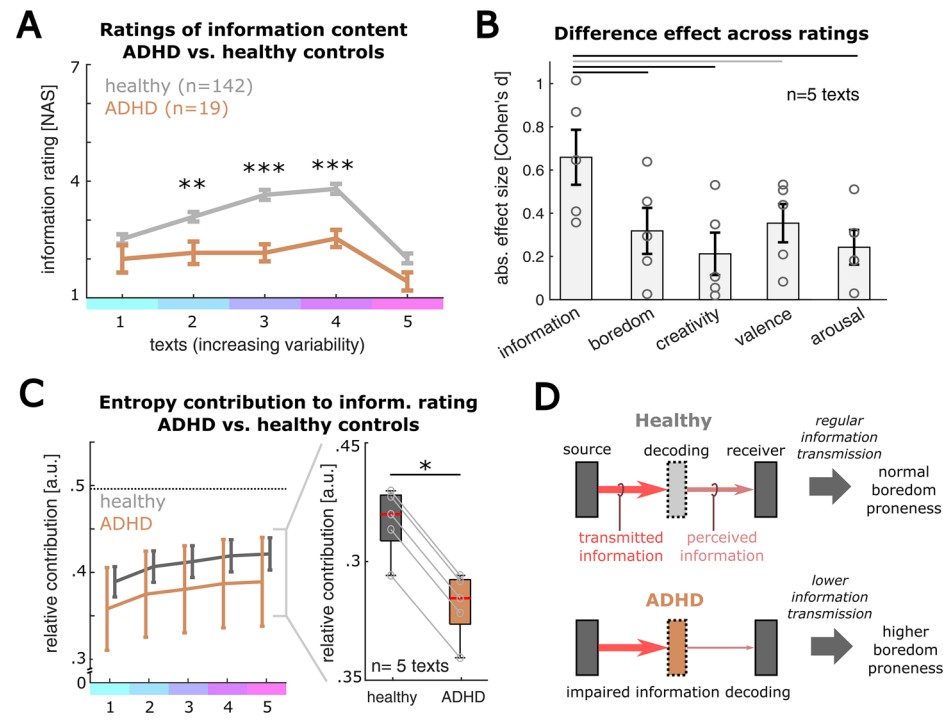

**Fig. 5 | Patients with attention deficit hyperactivity disorder show a reduced sensitivity to external information. A** In an equivalent experiment, adult outpatients with attention deficit hyperactivity disorder (ADHD, $n = 19$, see Methods) show a reduced perception of information for the texts, especially in regimes of moderate complexity (**$p < 0.01$, ***$p < 0.001$ in Wilcoxon rank sum test against 142 healthy control participants). Gray line indicates mean of healthy cohort, brown line indicates mean of ADHD cohort. Vertical bars indicate SEM. **B** Comparison of the size of the difference effects in the text ratings between ADHD patients and control participants. Each dot represents the difference effect for a given text. The bars and vertical error bars indicate the mean +-SEM. Perceived information shows the largest difference effect, compared to the other text sentiments (black lines indicate $p < 0.05$ in Wilcoxon rank sum test, gray line indicates $p > 0.05$). **C** Left: Comparison of the mean contribution of entropy to the perception of information over texts (reflecting individuals' sensitivity to retrieve external information) between control participants (gray, equivalent to Fig. 4D) and ADHD patients (brown). Lines indicate the means, bars indicate SEM. Boxplots display the median (red), the quartiles around the median (box) and the top and bottom quartile (whiskers). Right: comparison of the mean entropy contribution over all 5 texts. ADHD patients show reduced average sensitivity to external information as compared to healthy controls (*$p < 0.05$ in a signed rank test). **D** Summary schematic, depicting the reduced information transmission from external stimuli to ADHD patients, likely due to lower decoding abilities.

input. This assumes that the ability to retrieve information is not only determined by environmental sources of stimuli that send out information, but also on an individual's ability to process and decode the sent information and integrate it into pre-existing knowledge[47–49]. Our data provides empirical evidence for the crucial role of individual factors for information-decoding, where statistical modeling of perceived information revealed a central role of both, external and internal factors. Although our model only covers a limited number of parameters, focusing on entropy and the Big Five personality traits, it already explained a significant fraction of variance in perceived information, thus allowing a comparison of the relative contribution of internal and external factors on information transmission.

On the side of external factors of information transmission, intermediate levels of environmental complexity are considered as the best condition to effectively engage the own cognitive resources[71] and process incoming information effectively[70,72]. In line with this, we observed an inverse U-shaped trend of perceived information over the different text stimuli, which however was not entirely congruent with the participants' boredom ratings. In particular, we found that the text stimulus with highest randomness (text 5) was rated as uninformative and aversive, but only as hardly boring, highly arousing and creative at the same time. This counterintuitive result suggests that indeed participants do not retrieve relevant information from a highly random stimulus, but can compensate this lack of external information with internal processes that prevent boredom. Such internally generated information might amongst others result from spontaneous associative processes and mind-wandering during the perception of the stimuli[73–76], forming internal linkages to complement the sparse external

information. In addition, as text 5 despite its random semantic content still contained coherent syntactic and grammatical structure, it might have carried information on a different level. For instance, phonetic information of a text may likely be transmitted independently from semantic information, potentially explaining why the random text 5 was rated as semantically uninformative but hardly boring. For future studies, entirely random combinations of letters that cannot be interpreted in any meaningful way, could be used to implement textual stimuli with even more extreme entropy, potentially leading to higher boredom.

On the side of internal factors, we found a significant contribution of the Big Five personality traits on information transmission, in particular for trait openness. This accords with prior studies demonstrating a link of openness and appreciation of stimulus complexity[77–79]. Interestingly, trait openness has also been linked with creativity[80,81] and enhanced cognitive performance[82], suggesting that the perception of information is associated with the ability to employ it to form innovative associations and behaviors[83].

### Reduced information transmission in ADHD
In addition to our investigations in healthy control individuals, we tested information transmission, boredom and text sentiment in ADHD patients, known to exhibit chronically increased boredom susceptibility[42]. Given the framework of our study, characterizing boredom as a consequence of low information, this elevated boredom proneness in ADHD could result from two different cognitive mechanisms: As a first scenario, the transmission of external information to ADHD patients could be normal, but the patients tend to map particular degrees of information to generally higher degrees of

boredom. As a second scenario, the transmission of sensory information per se could be reduced, leading to lower levels of perceived information, driving boredom. Assuming the first scenario, we would expect that information ratings in our study should be similar across ADHD patients and control participants, whereas boredom should be significantly increased. In case of the second scenario, we would expect to find pronounced differences in perceived information content without strong differences in stimulus sentiment.

Interestingly, our study provides clear evidence for the latter hypothesis: Perceived information was the only feature significantly reduced in ADHD patients, whereas boredom, affect, arousal and perceived creativity of the stimuli showed widely comparable levels across cohorts (see Fig. 5). This suggests that difficulties to decode and retrieve external stimulation, resulting from lower information transmission in a given environment, constitutes a core cognitive factor explaining increased levels of boredom susceptibility. Supporting this assumption, our regression analysis demonstrated lower decoding abilities for ADHD patients across all text stimuli, which can be interpreted as a correlate of attentional deficits and high impulsivity[23,36,39], critically limiting the amount of information drawn from a sensory stimulus[23,36,39]. The reduced sensitivity to entropy of external stimuli is in line with prior studies that found altered and irregular patterns of sensory processing in ADHD[84–87], suggesting that ADHD is associated with less successful information-processing on a sensory level. This view suggests that higher levels of boredom in ADHD patients may be caused by situation-specific disturbances of information flow, instead of reflecting generally increased boredom proneness.

### Candidate neural correlates of information-extraction

While the specific cognitive mechanisms leading to lower information sensitivity remain to be identified and are likely multifaceted, our approach yields a general experimental framework to quantify these mechanisms across individuals. In particular, our results suggest that boredom arises as a core correlate of low information, affecting the behavioral responses to a given sensory input[15,21] and thus safeguarding a continuous transmission of relevant information[16,17]. Besides these behavioral implications, boredom could also enhance the efficiency of information processing on a neuronal level: By driving an individual's attention away from monotonous and towards informative sensory input, boredom could facilitate the generation of efficient neural codes, a principle found throughout our nervous systems[88–92]. In this regard, recent studies have linked boredom experience with specific activity patterns of the anterior insula[93,94], a brain region involved in the processing of stimulus salience[95–97], a concept closely related to relevance and information content. Thus, on an implementational level, the neuronal salience detection network could serve as a machinery to estimate and extract information from a given sensory input and drive corresponding cognitive responses, such as boredom, in order to ensure an optimal stream of information to the brain.

### Limitations

In our study, we estimate objective information content in the texts as empirical entropy, applying this framework to compare information processing in healthy control participants and ADHD outpatients. Even though we find that the entropy metric relates to other standard metrics of semantic variety and linguistic complexity[46,98,99], additional information might be conveyed on a symbolic or phonetic level, not covered by our approach[100].

This notion particularly concerns the ratings we acquired for the semantically random text 5, which in many was rated differently than the other texts in respect to affect and arousal. This text was intentionally designed with random semantic content but comparable syntactic structure, to implement higher entropy while maintaining the general stimulus structure. While this change of semantic content could potentially introduce covariates in the perception of text 5, we found that in general, text 5 was rated in a similar range as the other text stimuli. Moreover, we observed that the creativity rating for text 5 was linearly aligned with the creativity ratings

for the other texts, suggesting a valid extrapolation of stimulus complexity[61]. Further studies could thus aim to develop complementary stimulus sets with simpler dimensionality that are also controllable in their objective information content, but do not depend on language[16,101–104]. Such language-free stimulus sets with defined information content could be combined with diverse methodologies beyond self-reports to explore the neural, behavioral and bodily correlates of perceived information in more depth[16,20]. In addition, as our regression approach focuses on assessing the unique contributions of different personality traits on individual transmission of environmental information, future studies could complement our findings by investigating interactive effects of different internal features on decoding abilities.

Although we ensure coarse demographic similarity between the cohort of healthy control participants and adult ADHD patients in our study, we emphasize significant differences in the size and in the psychometric properties of our samples, limiting our sensitivity to resolve inter-individual differences in information-processing and the generalizability of our findings.

### Conclusion

Together, our study highlights that the amount of information perceived by the brain does not only depend on the information content of a given sensory input, but also critically depends on internal, trait-based factors that determine an individual's ability to decode external stimuli. Low information transmission is associated with elevated boredom experience, where reduced sensitivity to external stimuli predisposes individuals to experience higher boredom. In particular, in persons with attentional disorder, individual decoding deficits are aggravated, accounting for increased boredom susceptibility. Thus, we demonstrate that lower information transmission, with its internal and external underpinnings as central correlates of individual boredom experience, constitutes a potential target for specific cognitive interventions to enhance information flow and alleviate boredom.

### Data availability

The data of this study and the text stimuli used in this study are available via https://doi.org/10.12751/g-node.blnri7, or upon request from the corresponding authors.

### Code availability

The MATLAB code used to analyze the data of this study is available via https://doi.org/10.12751/g-node.blnri7, or upon request from the corresponding authors.

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

## Acknowledgements

We thank all the participants and patients who volunteered to contribute to this study. Moreover, we thank the team from the Mainz Behavioral and Experimental Laboratory (MABELLA) and the Neuroimaging Center (NIC) Mainz for their support during the preparation of the study. We also thank the

colleagues from the Department for Psychiatry Mainz for the support in the recruitment of ADHD patients. This work was supported by research grant Deutsche Forschungsgemeinschaft CRC1080-C05 (SR), Deutsche Forschungsgemeinschaft SPP 2041 Project #347573108 (SR), Deutsche Forschungsgemeinschaft/Agence nationale de la recherche Project #431393205 (SR), Deutsche Forschungsgemeinschaft DIP "Neurobiology of Forgetting" (JE, MK, SR), Rhine-Main University Alliance RMU (SR, MK) and a Focus Program Translational Neuroscience Mainz fellowship (JS). The funders had no role in study design, data collection and analysis, decision to publish or preparation of the manuscript.

## Author contributions

J.S. designed the study. J.S., O.T. and S.R. requested permission for the study from the local ethics committee. O.T., M.K., and S.R. jointly supervised the study. A.G. and J.S. recruited and tested the healthy control participants. A.K., J.S., and D.T. recruited and tested the ADHD patients. J.S., J.E., and V.M. analyzed the data. J.S. wrote the first draft of the manuscript. All authors edited the manuscript.

## Funding

## Competing interests

The authors declare no competing interests.
