## [Transparent Peer Review file · Communications Psychology]

A reduced perception of sensory information is linked with elevated boredom in mental health and attention-deficit hyperactivity disorder

Corresponding Author: Dr Johannes Seiler

Version 0:

Decision Letter:

Dear Dr Seiler,

Thank you for submitting your manuscript titled "A reduced perception of sensory information is linked with elevated boredom in health and attention-deficit hyperactivity disorder" to Communications Psychology.

We have given the paper our careful consideration and find it of potential interest. However, due to certain shortcomings we are concerned that sending the current manuscript out to review could lead to unnecessary delays and quite possibly an undesirable outcome of the review process.

In particular, all statements or interpretations of your results must be supported by appropriate, fully reported statistics. Please refer to our specific guidance below: <https://www.nature.com/commspsychol/submission-guidelines#statistical-guidelines>

It is generally against our guidelines to only convey results via Display items. Full statistics for each comparison must be redundantly reported in the text.

For manuscripts that interpret null results, we require Bayes Factors or equivalence tests to interpret the null results.

Please also ensure to use appropriate language to describe the results. Statements such as 'There is no difference between x and y.' or 'X does not affect Y.' must be revised to read 'We found [no/little] credible evidence of a difference between x and y.' or 'We found [no/little] credible evidence that X affects Y.'. This point is particularly pertinent for your work where you draw inferences about the absence of group differences, for example:

"we did not find relevant interactions between the text ratings and potentially confounding psychometric features of the participants, underlining the high degree of homogeneity in our healthy sample (Supplementary Figure 1B)." [also note that full statistics need to be reported in the text, not only a Figure in the SI]

We would therefore like to invite you to revise your manuscript to address these concerns before we make a final determination on whether to send your manuscript for external review.

We shall hope to receive your revised version as soon as you are able to complete the suggested revisions. If something similar is published in the interim we will have to consider the impact it has on the novelty of a revised manuscript.

If you anticipate a delay of more than four weeks, please let us know. Should your manuscript be substantially delayed without notifying us in advance and your article is eventually published, the received date may be that of the revised, not the original, version.

We also ask that you ensure your manuscript complies with our editorial policies and reporting requirements.

To that end, we require revised manuscripts to be accompanied by two completed items: a reporting summary that collects information on study design and procedure, and an editorial policy checklist that verifies compliance with all required editorial policies.

- <https://www.nature.com/documents/nr-reporting-summary.zip>>Nature Research Reporting Summary
- <https://www.nature.com/documents/nr-editorial-policy-checklist.pdf>>Editorial Policy Checklist

All points on the policy checklist must be addressed. Your revised manuscript can only be sent to referees if these checklists are completed and uploaded with the revision.

If you are not interested in submitting a suitably revised manuscript in the future please let me know immediately so we can close your file. If you have any questions, please contact me.

Please use the link below when you are prepared to resubmit.
Link Redacted

Thank you for your interest in Communications Psychology.

Best regards,
Neil Garrett

Neil Garrett, PhD
Editorial Board Member
Communications Psychology
orcid.org/0000-0003-1440-472X

Version 1:

Decision Letter:

Dear Dr Seiler,

Thank you for your patience during the peer-review process. Your manuscript titled "A reduced perception of sensory information is linked with elevated boredom in health and attention-deficit hyperactivity disorder" has now been seen by 2 reviewers, and I include their comments at the end of this message. They find your work of interest but raised some important points. We are interested in the possibility of publishing your study in Communications Psychology, but would like to consider your responses to these concerns and assess a revised manuscript before we make a final decision on publication.

We therefore invite you to revise and resubmit your manuscript, along with a point-by-point response to the reviewers. Please highlight all changes in the manuscript text file.

Editorially, we consider each of the points raised by the reviewers to be of importance. But in particular:

Both Reviewers urge more careful consideration of the stimuli used in the task. This includes how these stimuli are interpreted (difficulty vs entropy) given the ratings (R1). Particular attention is needed to Text 5 in particular. R1 asks if this stimulus was potentially not made difficult/random enough which could explain why the boredom ratings do not assume a U shape (which would be more in line with past findings. Note: this has implications for how to interpret group differences). R2 asks whether there is a confound introduced by text 5 by virtue of the fact this has words/content unrelated to texts 1-4 - could this explain the fact that this emerges as an outlier in the models? Is treating text complexity as a linear variable appropriate? Please consider ways to address this in the analysis.

R2 draws attention to the fact that the negative relationship between information sensitivity and boredom is only reported in the healthy sample. Please report the relationship between information sensitivity and boredom in the ADHD groups.

In terms of the modelling, consider why internal factor contributions exceed external factors across text complexities and why external and internal factors contribute independently, not interactively (R2). Can the model tell us something about the direction of the effects of each factor (R1)? This would be more informative than simply reporting whether factors have a high or low contribution on information processing).

In addition, please ensure you follow our statistical guidelines when reporting statistics (<https://www.nature.com/commspsychol/submit/submission-guidelines#statistical-guidelines>). Please note in particular our requirements for the reporting and interpretation of null-results. Non-significant findings derived from null-hypotheses significance tests should be reported in full, but may not be interpreted. Where you interpret null results, this interpretation

must be based on Bayes Factors or equivalence tests. ****Compliance with statistical reporting requirements is mandatory at this stage and a precondition for re-review.****

I am attaching an Editorial Requests Table that details critical reporting requirements for the revised manuscript. Please attend to each item and ensure your manuscript is fully compliant. If your revised manuscript is not aligned with these requests on major issues, such as those concerning statistics, it may be returned to you for further revisions without re-review.

Please submit the following items:

- Revised manuscript
- Point-by-point response to the referees' comments
- Cover letter (as a separate document)
- [Nature Research Reporting Summary](https://www.nature.com/documents/nr-reporting-summary.zip)
- [Editorial Policy Checklist](https://www.nature.com/documents/nr-editorial-policy-checklist.pdf)
- Completed Editorial Request Table (attached).

via this link: Link Redacted .

Additional guidance is available in our style and formatting guide [Communications Psychology formatting guide](https://www.nature.com/documents/commpsychol-style-formatting-guide-accept.pdf).

Best regards,

Neil Garrett

Neil Garrett, PhD
Editorial Board Member
Communications Psychology
orcid.org/0000-0003-1440-472X

REVIEWER EXPERTISE:

Reviewer #1: Boredom, Modelling of behavioural data

Reviewer #2: Boredom, Modelling of behavioural data

REVIEWER REPORTS:

Reviewer #1 (Remarks to the Author):

The authors explored the link between information processing and boredom within a framework suggesting that individual differences in information processing might underly differences in susceptibility to boredom. They did so with stimuli defined

by distinct levels of entropy and examined the role of ADHD in extracting information. Results showed that boredom ratings tracked with entropy (less boredom with higher entropy) and that adults with ADHD extracted less information from the texts used than did neurotypical controls.

This work casts boredom and boredom proneness within an information processing framework and has developed a clever way of testing this, following on from earlier work in this group. I think the work is of great value to those interested in boredom and is well worth publishing here. My comments below are merely intended as suggestions that the authors can take or leave.

The inverted-U shape for the complexity ratings mirrors what Ulrich and colleagues (cited by the authors) showed in neuroimaging data when all they manipulated was task difficulty (they were looking at flow states). I wonder if the authors could address the possibility that this is what is going on here too - entropy is what is quantitatively measured and manipulated, but difficulty is what is felt by the participants - too much entropy is too difficult to parse? I don't think this changes the interpretation much, just couches it slightly differently.

In a similar vein, I was pleased to see the work of Orin Klapp cited, but his account, and the one given here, are essentially meaning-based accounts of boredom when you boil it down. Parsing signal from noise (manipulated by the objective entropy) is vital in establishing meaning. Making that task difficult makes it meaningless. I'd be interested in the authors' thoughts on this.

Also, the accounts of Westgate and Wilson would suggest (and show) a kind of U-shape function to boredom with respect to effort - so boredom is highest at the low and high end of rated effort/task difficulty. The text 5 results here contradict that work (and work by Struck et al on control which mirrors the Westgate findings). It is plausible I guess that text 5 didn't have "enough" randomness. My reading of the example in Figure 1 was that the "story" was weird but comprehensible. I can imagine giving participants some abstruse philosophy text might be harder to engage with than what was used here. Again, I think this is a minor point, but worth attempting to reconcile given that Westgate and Wilson, (and Danckert & Elpidorou's 2023 paper) accounts, and Klapp's account, suggest boredom should be highest at the low and high ends of entropy - a U-shaped function not quite seen here. The fact that text 5 was rated to be highly creative bolsters my point I think - while it reads oddly, many might think "Well, you know those creative types, always writing weird stuff!" and so not rate it as boring. Also seems that the bored ratings bottom out at text 4 and 5 - so I imagine them going back up if you could somehow make an even more random text 6.

The above point I think is also particularly relevant for the ADHD work that comes later. Is it that this patient group (with a relatively small n here) are less capable of processing the information, or do they find less meaning in it and so disengage? I'm not sure the current work can disambiguate those two possibilities. Again, I don't see this as a flaw in the work, just something worth highlighting as a potential limitation to the interpretations given.

When training the model on external and internal factors - why choose only the Big 5 personality factors for the latter? Why not choose boredom proneness given the focus of the paper? And for that matter, other factors related consistently with boredom proneness like self-control? Seemed like an odd omission from the model.

Also, one would not expect that each of the Big 5 traits would function in a uniform way - openness to experience and neuroticism might be expected to affect information processing in opposing directions. Can the model tease out these kinds of hypotheses? I see that the authors talk of "higher" and "lower" contributions - but this is not directional. Can the model tell us something about directional effects (e.g., openness leads to better information processing, neuroticism to worse?).

Reviewer #2 (Remarks to the Author):

The manuscript "A reduced perception of sensory information is linked with elevated boredom in health and attention-deficit hyperactivity disorder" examined the external and internal factors influencing information transmission and its association with boredom in healthy and ADHD samples. The study addressed an important research question with an innovative design of text stimuli and quantifying text complexity using metrics like entropy and word embeddings. While this paper presents valuable findings, I have several suggestions for improvement.

The introduction was a bit difficult to follow without reading the method section later. Terms like "sensory information" (state of information provided by the environment) and "information transmission" (process of information received by an individual) were not clearly defined and distinguished early on, which might have confused external factors with the decoding process. While the existing evidence centered on low sensory information as a source of boredom, the focus shifted abruptly to information transmission without a proper transition in line 60, leaving the theoretical foundation unclear. Defining key terms early and ensuring smoother transitions would improve readability and coherence.

The methods demonstrated strengths in design, leveraging text entropy and word embeddings to quantify external factors of information transmission. These methodological approaches were novel and contributed significantly to the field. However,

there were issues such as reference errors. For instance, Supplementary Table 3 was incorrectly cited as the final version of the text stimuli in line 387 when it actually described demographic characteristics. Additionally, the design of text 5 might introduce covariates beyond text complexity by using unrelated words to text 1-4, which was not explained and controlled in the analyses. The imbalanced sample sizes between healthy participants (N=142) and ADHD patients (N=19) also raised concerns about the validity of group comparisons.

In the results section, the authors first reported the associations between objective text complexity and subjective ratings regarding perception, affect, boredom, and creativity. However, text 5 emerged as an "outlier" in perception and boredom models, probably due to its unrelated content to text 1-4, which made me wonder whether it was appropriate to treat text 1-5 as a linear representation of text complexity. The authors did not provide the rationale for why they twisted the tone for text 5 and how to tackle the potential covariates it might introduce. Furthermore, data revealed a negative relationship between information sensitivity and boredom, which adds depth to the understanding of sensory information processing. However, when the manuscript title claimed this negative relationship in both healthy and ADHD samples, it was only reported in the healthy sample, not for the ADHD sample, leaving a critical gap in understanding the ADHD-related findings.

The discussion reiterated the link between information sensitivity and boredom. However, this result is not novel, as prior research (e.g., Westgate & Wilson, 2018) has demonstrated that people are more likely to experience boredom in overstimulated and under-stimulated environments. The manuscript would benefit from a clearer articulation of how its findings extend existing knowledge. The novel contribution of quantifying text complexity and its psychophysical framework is noteworthy. Additionally, the authors did not address why internal factors' contributions exceed external factors across text complexities. Also, external and internal factors seem to contribute independently, not interactively. Expanding on these points and discussing limitations more explicitly would strengthen the manuscript.

If you experience problems in linking your ORCID, please contact the Platform Support Helpdesk.

Version 2:

Decision Letter:

Dear Dr Seiler,

Your manuscript titled "A reduced perception of sensory information is linked with elevated boredom in health and attention-deficit hyperactivity disorder" has now been seen by our reviewers, whose comments appear below. In light of their advice I am delighted to say that we are happy, in principle, to publish a suitably revised version in Communications Psychology.

We therefore invite you to revise your paper one last time to address the remaining concerns of our reviewers and a list of editorial requests. At the same time we ask that you edit your manuscript to comply with our format requirements and to maximise the accessibility and therefore the impact of your work.

EDITORIAL REQUESTS:

SUBMISSION INFORMATION:

OPEN ACCESS:

* DATA AVAILABILITY:

Link Redacted

Best regards,

Jennifer Bellingtier

Jennifer Bellingtier, PhD
Senior Editor
Communications Psychology

Neil Garrett, PhD
Editorial Board Member
Communications Psychology
orcid.org/0000-0003-1440-472X

REVIEWER EXPERTISE:

Reviewer #1: Boredom, Modelling of behavioural data

REVIEWERS' COMMENTS:

Reviewer #1 (Remarks to the Author):

The authors have done a good job addressing my concerns. I think the paper is ready for publication now.

To the
Editorial Board of
Communications Psychology

Research Group for Systemic Neurophysiology

Dr. med. Johannes P.H. Seiler
Group of Prof. S. Rumpel for Systemic Neurophysiology
Hanns-Dieter-Hüsch-Weg 19
55128 Mainz, Germany
Phone: +49 (0) 6131 39-27356
Fax: +49 (0) 6131 39-26071
www.unimedizin-mainz.de/physiologie/ag-prof-simon-rumpel

Mainz, 08/02/2025

Point-by-point response to reviewers' comments on the manuscript "A reduced perception of sensory information is linked with elevated boredom in health and attention-deficit hyperactivity disorder"

All authors would like to thank the reviewers for their time and constructive criticism. These comments have led to substantial adjustments of the manuscript, in our opinion, significantly improving its quality. The respective adjustments with regards to the content of the manuscript are marked with yellow background color in the attached document.

In the following, we address the reviews point by point:

Reviewer #1 (Remarks to the Author):

The authors explored the link between information processing and boredom within a framework suggesting that individual differences in information processing might underly differences in susceptibility to boredom. They did so with stimuli defined by distinct levels of entropy and examined the role of ADHD in extracting information. Results showed that boredom ratings tracked with entropy (less boredom with higher entropy) and that adults with ADHD extracted less information from the texts used than did neurotypical controls.

This work casts boredom and boredom proneness within an information processing framework and has developed a clever way of testing this, following on from earlier work in this group. I think the work is of great value to those interested in boredom and is well worth publishing here. My comments below are merely intended as suggestions that the authors can take or leave.

Thank you.

The inverted-U shape for the complexity ratings mirrors what Ulrich and colleagues (cited by the authors) showed in neuroimaging data when all they manipulated was task difficulty (they were looking at flow states). I wonder if the authors could address the possibility that this is what is going on here too - entropy is what is quantitatively measured and manipulated, but difficulty is what is felt by the participants - too much entropy is too difficult to parse? I don't think this changes the interpretation much, just couches it slightly differently.

Thank you for this interesting comment. We agree that the study of Ulrich et al., manipulating the degree of task difficulty, is conceptually related to our manipulation of entropy. However, in our case the text

stimuli were presented passively without requiring any action from the subjects, different from Ulrich et al., where subjects had to conduct simple mathematical computations. Entropy in the context of our study describes the mere structure of the text stimuli presented to participants, where the mathematical tasks constitute problems that require participants' knowledge and skills to solve them. Thus, when mapping our experiment to an axis between the action-related concept of 'difficulty' on the one side, and the more passive concept of 'entropy' on the other side, I would expect that our text ratings reflect mostly passive aspects of perceptual complexity rather than difficulty.

To give an analogy, I would compare the situation in our task with a visit in a museum for modern art: Although some pieces of art that provide very high entropy may induce confusion and dislike, they are still not experienced as 'difficult', as they do not require subjects to deeply engage with them, or to understand them.

We believe, the thought by the reviewer is very interesting and have therefore revised parts of our Discussion section to address this aspect (see l. 455 ff.).

In a similar vein, I was pleased to see the work of Orin Klapp cited, but his account, and the one given here, are essentially meaning-based accounts of boredom when you boil it down. Parsing signal from noise (manipulated by the objective entropy) is vital in establishing meaning. Making that task difficult makes it meaningless. I'd be interested in the authors' thoughts on this.

Definitely, the extraction of information from a noisy input is a crucial component to perceive a stimulus as meaningful. Nevertheless, I would also argue that the meaning provided by a stimulus is not linked to its entropy per se, but rather to the topic of conveyed information, relative to an individual's current demands. For instances, while information about a novel food source could have very high meaning but only provides low overall entropy, a telephone book may contain a high amount of information that however is meaningless. Similarly, I assume a task can be meaningful although being difficult and mentally effortful. Thus, meaning can be one of the reasons why humans actively seek mental effort (Clay et al., 2022).

In the framework of our study, I would see decoding of information as a first step, to extract *potential meaning* from sensory inputs. The *actual meaning* of a sensory input will be decided in a second step, where the incoming information is interpreted relative to preexisting knowledge and the current demands of an individual. Thus, information can be useful and meaningful whenever it fills a gap in individual knowledge networks (Loewenstein, 1994, Seiler and Dan, 2024).

We thank the reviewer for this insightful aspect and have added a paragraph to our Discussion section, discussing the relationship of information-processing and meaning. (l. 464 ff.)

Also, the accounts of Westgate and Wilson would suggest (and show) a kind of U-shape function to boredom with respect to effort - so boredom is highest at the low and high end of rated effort/task difficulty. The text 5 results here contradict that work (and work by Struk et al on control which mirrors the Westgate findings). It is plausible I guess that text 5 didn't have "enough" randomness. My reading of the example in Figure 1 was that the "story" was weird but comprehensible. I can imagine giving participants some abstruse philosophy text might be harder to engage with than what was used here. Again, I think this is a minor point, but worth attempting to reconcile given that Westgate and Wilson, (and Danckert & Elpidorou's 2023 paper) accounts, and Klapp's account, suggest boredom should be highest at the low and high ends of entropy - a U-shaped function not quite seen here. The fact that text 5 was rated to be highly creative bolsters my point I think - while it reads oddly, many might think "Well, you know those creative types, always writing weird stuff!" and so not rate it as boring. Also seems that the bored ratings bottom

out at text 4 and 5 - so I imagine them going back up if you could somehow make an even more random text 6.

We thank the reviewer for pointing out the expected U-shaped function of boredom over effort and task complexity. Indeed, we were also expecting that boredom, just as subjective information, would show such a U-shaped relationship with the entropy of our texts. Surprisingly, although perceived information did show an inverse U-shaped relation, boredom ratings did not. We agree with the reviewer that this could likely be due to some remaining comprehensibility in the text which, although the semantic meaning was highly incoherent, still had syntactical structure. Following this idea, there are likely different types of information transmitted by the text: In our approach, we focus on quantifying the semantic entropy of texts, however, one could also expect that phonetic, graphical or syntactical features of the texts convey different types of information, which we do not capture in our study design. These non-semantic types of information could contribute to the fact that text 5 is not rated as highly boring. Maybe, if one would have tested a completely random combination of letters, one could find that boredom increases again.

We believe that this point, raised by the reviewer, is very important. In order to address it, we have expanded parts of our Discussion section to elaborate on this aspect (see l. 490 ff., 551 ff.).

The above point I think is also particularly relevant for the ADHD work that comes later. Is it that this patient group (with a relatively small n here) are less capable of processing the information, or do they find less meaning in it and so disengage? I'm not sure the current work can disambiguate those two possibilities. Again, I don't see this as a flaw in the work, just something worth highlighting as a potential limitation to the interpretations given.

It is true that our approach quantifies information transmission indirectly, by regressing perceived information ratings from all texts with text entropy and internal personality features. We further agree with the reviewer the deficits in information sensitivity found in ADHD patients, could arise on different levels of processing: either on a lower hierarchical level of sensory processing (by the reviewer phrased as “processing the information”), or on a higher hierarchical level of processing, cognitively interpreting the sensory input (by the reviewer termed phrased as “finding meaning”). Although, we cannot directly assess these two factors, given the simple psychometric assessments in our study, our statistical modeling allows to indirectly address them: Our model allows to disambiguate between *external* and *internal* factors of information perception. Likely, our external factors reflect sensory features of the stimulus-processing on a low hierarchical level (e.g. salience effects described in Berlyne, 1966 or Bender et al., 2017), whereas, our internal factors covering Big Five personality features reflect processes on higher cognitive levels of processing (Lionetti et al., 2019).

Our observation, that in ADHD patients primarily the external factor of information-processing, expressed as a sensitivity to stimulus entropy, is reduced (see Figure 5C), suggests that ADHD have deficits in information-processing on a lower, sensory level.

To address the reviewer’s comment, we have added a note on this point to the Discussion section of our revised manuscript (see l. 523 ff.)

When training the model on external and internal factors - why choose only the Big 5 personality factors for the latter? Why not choose boredom proneness given the focus of the paper? And for that matter, other factors related consistently with boredom proneness like self-control? Seemed like an odd omission from the model.

Thank you for this comment. While we agree that it appears intuitive to assess boredom proneness as a predictor for information perception, we intentionally decided against this. Our reason to exclude boredom proneness from our model was to regress perceived information by a naive set of general personality features (the Big Five), that do not have an inherent link to boredom. This approach had the advantage that it allowed in a second step to assess the relationship between the parameter contributions derived from the model and the boredom elicited by each text (see Figure 5F). Here, if boredom proneness would have been included as a predictor, any link between parameter contributions and the rated boredom for each text would have been expected by construction. Instead, the fact that our model, including only general personality features and entropy, allows to predict subjects' experienced boredom based on the sensitivity of subjects to entropy (Figure 5F), underlines that the sensitivity to incoming information directly relates to the degree of boredom elicited by an incoming stimulus.

To address the reviewer's point and describe our approach clearly, we included an elaborate description of our rationale in the revised manuscript (see l. 345 ff.).

Also, one would not expect that each of the Big 5 traits would function in a uniform way - openness to experience and neuroticism might be expected to affect information processing in opposing directions. Can the model tease out these kinds of hypotheses? I see that the authors talk of "higher" and "lower" contributions - but this is not directional. Can the model tell us something about directional effects (e.g., openness leads to better information processing, neuroticism to worse?).

In our analysis of parameter contributions, we indeed only quantify the general impact that a given parameter (one of the Big Five personality features) has on the ratings of information content. We agree with the reviewer that it could be interesting to further depict the kind of relationship (positive or negative) of each parameter with information ratings. Quantifying the pairwise correlation between each parameter and information ratings provides a measure that allows to estimate this kind of relationship. In the previous version of our manuscript, we only presented these correlations between the Big Five traits and information ratings in Supplementary Figure 1B, without specific emphasis. We are thankful for the reviewer's comment pointing out the relevance of this analysis for the interpretation of our model and therefore have added a display of the correlations between model parameters and information rating, next to the contribution analysis in Figure 4E (see also revised Results section l. 365 ff.).

Reviewer #2 (Remarks to the Author):

The manuscript "A reduced perception of sensory information is linked with elevated boredom in health and attention-deficit hyperactivity disorder" examined the external and internal factors influencing information transmission and its association with boredom in healthy and ADHD samples. The study addressed an important research question with an innovative design of text stimuli and quantifying text complexity using metrics like entropy and word embeddings. While this paper presents valuable findings, I have several suggestions for improvement.

Thank you for appreciating the approach in our study.

The introduction was a bit difficult to follow without reading the method section later. Terms like "sensory information" (state of information provided by the environment) and "information

transmission” (process of information received by an individual) were not clearly defined and distinguished early on, which might have confused external factors with the decoding process. While the existing evidence centered on low sensory information as a source of boredom, the focus shifted abruptly to information transmission without a proper transition in line 60, leaving the theoretical foundation unclear. Defining key terms early and ensuring smoother transitions would improve readability and coherence.

We apologize if our reasoning in the introduction was unclear. In order to address this comment, we have revised our Introduction section to describe more clearly which concepts we build our study on (l. 41 ff., 63 ff., 70 ff.). In particular, we emphasized concepts of information theory early on, pointing out specific links to boredom, as described in a recent study (Seiler and Dan, 2024). We hope, these adjustments increase the comprehensibility of our introduction.

The methods demonstrated strengths in design, leveraging text entropy and word embeddings to quantify external factors of information transmission. These methodological approaches were novel and contributed significantly to the field. However, there were issues such as reference errors. For instance, Supplementary Table 3 was incorrectly cited as the final version of the text stimuli in line 387 when it actually described demographic characteristics.

Thank you for the careful reading. We corrected our reference (referring to Supplementary Figure 2 in the revised manuscript).

Additionally, the design of text 5 might introduce covariates beyond text complexity by using unrelated words to text 1-4, which was not explained and controlled in the analyses.

In the results section, the authors first reported the associations between objective text complexity and subjective ratings regarding perception, affect, boredom, and creativity. However, text 5 emerged as an “outlier” in perception and boredom models, probably due to its unrelated content to text 1-4, which made me wonder whether it was appropriate to treat text 1-5 as a linear representation of text complexity. The authors did not provide the rationale for why they twisted the tone for text 5 and how to tackle the potential covariates it might introduce.

The purpose in designing text 5 with words unrelated from texts 1-4, was to create one textual stimulus with very high entropy and semantic complexity, in order to cover a broad spectrum of information content with our stimuli. Thus, it was our explicit aim to create a stimulus that deviates from the other ones by having higher complexity. However, we agree with the reviewer that this design could potentially lead to different dimensions of information (e.g. different semantic content, or different phonetic information, etc.) being conveyed by text 5, leading to a variation of complexity across texts which not necessarily linear (see also response to reviewer 1, rebuttal letter p. 2-3). The notion that text 5 was rated with high creativity, linearly following the other texts’ ratings, as well as with boredom and affect comparable to other text versions, in our view supports the assumption that the texts were comparable in many respects.

The reviewer’s comment made clear for us that this point warrants a deeper discussion in our paper. To address the criticism, we have therefore added a detailed discussion on the specifics of text 5 and its potential influence on the interpretation of our findings to our revised Discussion section (see p. 490 ff., 551 ff.).

The imbalanced sample sizes between healthy participants (N=142) and ADHD patients (N=19) also raised concerns about the validity of group comparisons.

To address this criticism, we have added a comment on the differences in sample sizes to our revised Discussion section (see l. 566 ff.). Furthermore, we are currently preparing a follow-up study to test entropy sensitivity in the context of information-processing and boredom with age-sex-matched ADHD patients and healthy control subjects, therefore providing enhanced comparability.

Furthermore, data revealed a negative relationship between information sensitivity and boredom, which adds depth to the understanding of sensory information processing. However, when the manuscript title claimed this negative relationship in both healthy and ADHD samples, it was only reported in the healthy sample, not for the ADHD sample, leaving a critical gap in understanding the ADHD-related findings.

Thank you very much for this valuable comment. We agree that replicating the correlation of information sensitivity and experienced boredom in our ADHD sample would potentially add to our findings. When, equivalently to the analysis with healthy subjects (see Figure 4F) testing the relationship between entropy sensitivity and mean boredom rating in the ADHD sample, we found no significant correlation: $n=19$ ADHD patients, $R=0.299$, $p=0.214$. The fact that this observation does not reach statistical significance may likely be due to the small size of our ADHD sample as well as to the overall reduced levels of entropy sensitivity in ADHD leading to lower inter-individual variance (see Figure 5C). In our view, a larger ADHD sample would be required to reliably resolve inter-individual differences within the patient group and test their potential mechanistic underpinnings.

To address the reviewer's criticism, we added this analysis to the Results section of our revised manuscript (see l. 421 ff.).

The discussion reiterated the link between information sensitivity and boredom. However, this result is not novel, as prior research (e.g., Westgate & Wilson, 2018) has demonstrated that people are more likely to experience boredom in overstimulated and under-stimulated environments. The manuscript would benefit from a clearer articulation of how its findings extend existing knowledge. The novel contribution of quantifying text complexity and its psychophysical framework is noteworthy.

This comment is similar to the point raised by reviewer 1 on p. 1-2). We agree with the reviewer that our findings are in line with other studies that have shown a U-shaped relation between boredom and task difficulty (Westgate and Wilson, 2018, Ulrich et al., 2014). However, our study significantly differs from these investigations in two main points: First, in our study boredom was tested for passively perceived stimuli that differed in entropy, whereas the reference studies used behavioral tasks that required engagement and active participation. Second, while the work of Westgate & Wilson or Ulrich et al. provided qualitative, ordinaly scaled descriptions of task complexity/difficulty, our study provides a novel parametric perspective on complexity, using information theory to quantify the conveyed information from a sensory stimulus.

In order to address the reviewer's comment, we expanded our Discussion section, elaborating on the conceptual relation to the mentioned studies and on the novel advancements provided by our quantitative approach (see l. 455 ff.).

Additionally, the authors did not address why internal factors' contributions exceed external factors across text complexities. Also, external and internal factors seem to contribute independently, not interactively. Expanding on these points and discussing limitations more explicitly would strengthen the manuscript.

Thank you for this comment. In our regression analysis, we use a lasso regularization to address the unique effects of the different model parameters to rated information (see l. 233). With this, our approach focuses indeed on the independent effects of entropy and personality traits on the perception of information. In order to provide further information on the direction in which the model parameters affect information perception, we have added correlation analyses to Figure 4E of our revised manuscript (see also response to the comment of reviewer 1 on p. 3-4). To further address the reviewer's point, we expanded our Discussion and Methods section, considering strengths and limitations of our regression approach in more detail (see l. 562 ff.).

Together, we hope that all these adjustments clarify the scientific contribution of our work, setting it into a meaningful context with prior research.

Sincerely,

Johannes Seiler (in the name of all authors)

References:

- BENDER, A. R., NAVEH-BENJAMIN, M., AMANN, K. & RAZ, N. 2017. The role of stimulus complexity and salience in memory for face–name associations in healthy adults: Friend or foe? *Psychology and Aging*, 32, 489-505.
- BERLYNE, D. E. 1966. Curiosity and Exploration. *Science*, 153, 25-33.
- CLAY, G., MLYNSKI, C., KORB, F. M., GOSCHKE, T. & JOB, V. 2022. Rewarding cognitive effort increases the intrinsic value of mental labor. *Proceedings of the National Academy of Sciences*, 119, e2111785119.
- LIONETTI, F., PASTORE, M., MOSCARDINO, U., NOCENTINI, A., PLUESS, K. & PLUESS, M. 2019. Sensory Processing Sensitivity and its association with personality traits and affect: A meta-analysis. *Journal of Research in Personality*, 81, 138-152.
- LOEWENSTEIN, G. 1994. The psychology of curiosity: A review and reinterpretation. *Psychological Bulletin*, 116, 75-98.
- SEILER, J. P. H. & DAN, O. 2024. Boredom and curiosity: the hunger and the appetite for information. *Frontiers in Psychology*, 15.
- ULRICH, M., KELLER, J., HOENIG, K., WALLER, C. & GRON, G. 2014. Neural correlates of experimentally induced flow experiences. *Neuroimage*, 86, 194-202.
- WESTGATE, E. C. & WILSON, T. D. 2018. Boring thoughts and bored minds: The MAC model of boredom and cognitive engagement. *Psychological Review*, 125, 689-713.